# Effects of sleep quality on non-alcoholic fatty liver disease: a cross-sectional survey

Atsushi Takahashi ![ORCID],[1] Yukio Anzai,[2] Masahito Kuroda,[3] Masae Kokubun,[4] Yuichiro Kondo,[5] Takashi Ogata,[6] Masashi Fujita,[1] Manabu Hayashi,[1] Hiromichi Imaizumi,[1] Kazumichi Abe,[1] Nobuo Tanji,[2] Hiromasa Ohira[1]

[1]Gastroenterology, Fukushima Medical University School of Medicine, Fukushima, Japan
[2]Gastroenterology, Watari Hospital, Fukushima, Japan
[3]Gastroenterology, Fukushima Red Cross Hospital, Fukushima, Japan
[4]Total Medical Checkup, Jusendo Hospital, Koriyama, Fukushima, Japan
[5]Gastroenterology, Fujita General Hospital, Date-gun, Fukushima, Japan
[6]Gastroenterology, Masu Memorial Hospital, Nihonmatsu, Japan

**Correspondence to**
Dr Atsushi Takahashi;
junior@fmu.ac.jp

## ABSTRACT

**Background** The effects of sleep quality on the risk of developing non-alcoholic fatty liver disease (NAFLD) remain uncertain. The purpose of this study was to clarify the association between sleep quality and NAFLD.

**Methods** The data of 4828 participants who underwent health check-ups at four hospitals were analysed. Sleep quality was evaluated by the Pittsburgh Sleep Quality Index (PSQI), which comprised seven elements scored from 0 to 3. The global PSQI score and the score for each element were compared between NAFLD and non-NAFLD groups separately by sex. Logistic regression analysis was performed to determine the association between NAFLD and each PSQI score.

**Results** In both men and women, the mean PSQI score for sleep medication use was significantly higher in non-NAFLD than in NAFLD. With regard to sleep medication use in men, the OR (95% CI) for NAFLD was lower with a score of 3 (OR 0.60, 95% CI 0.38–0.95) than with a score of 0 on multivariate logistic regression analysis adjusted for age, smoking habits and physical activity. The OR for NAFLD based on daytime dysfunction was also higher with a score of 3 than with a score of 0 in both men (OR 2.82, 95% CI 1.39–5.75) and women (OR 2.08, 95% CI 1.10–3.92). After adjustment for body mass index, the sleep latency scores in men and daytime dysfunction in women were associated with NAFLD.

**Conclusion** Sleep quality was associated with NAFLD, and there were sex differences.

## Strengths and limitations of this study

► The strength of this study lies in the large sample size from multiple centres.
► This is the first study to use all seven element scores of the Pittsburgh Sleep Quality Index to assess the association between sleep quality and non-alcoholic fatty liver disease (NAFLD).
► The present study was based on a self-reported questionnaire (Pittsburgh Sleep Quality Index); therefore, the associations between sleep quality and NAFLD need to be confirmed by objective methods.

## INTRODUCTION

Non-alcoholic fatty liver disease (NAFLD) is the most common chronic liver disease, probably because of the widespread prevalence of unhealthy lifestyle habits, such as excess nutritional intake or less physical activity.[1] In patients with NAFLD, liver fibrosis has been associated with liver-related outcomes and mortality.[2] Lifestyle interventions, such as diet modification and exercise to achieve weight loss, are fundamental in the treatment of patients with NAFLD; in fact, weight loss of more than 7%–10% has been reported to improve liver fibrosis in patients with NAFLD.[3–5] However, compared with nutrition or exercise, sleep disorder has attracted less attention. Of the several sleep disorders, obstructive sleep apnoea and short sleep duration have been established as risk factors for NAFLD.[6–11] On the other hand, only a few reports of the association between sleep quality and NAFLD have been published.[9 12 13]

The Pittsburgh Sleep Quality Index (PSQI) is an established and widely used method for sleep quality evaluation; it comprises seven elements, including subjective sleep quality, sleep latency, sleep duration, habitual sleep efficiency, sleep disturbance, sleep medication use and daytime dysfunction.[14] In previous reports of the sleep quality of patients with NAFLD, only the results of the global PSQI score or some components of the PSQI were assessed.[9 12 13] Therefore, the associations of all components of the PSQI with NAFLD are poorly understood. In addition to sleep time distribution,[8] the impact of sleep quality on NAFLD differs by sex.[9] In the present study, the aim was to clarify the association between sleep quality and NAFLD by analysing all elements of the PSQI separately by sex and by adjusting for confounding factors.

## METHODS

### Subjects

The subjects of this study were 6138 Japanese men and women who underwent health check-ups from May 2013 to March 2014 at the Watari Hospital Health Center, Jusendo Hospital, Fujita General Hospital, Fukushima Red Cross Hospital and Masu Memorial Hospital in Fukushima Prefecture, Japan. A total of 423 subjects who had insufficient data and 887 subjects who had chronic liver disease secondary to other causes (ie, hepatitis C antibody-positive, hepatitis B surface antigen-positive, and alcoholic consumption >30 g/day for men and >20 g/day for women) were excluded. NAFLD was defined as fatty liver detected by ultrasound in the absence of the above-mentioned causes of chronic liver disease based on an established guideline.[15] Accordingly, 4828 subjects (1864 men and 2964 women) were eligible for this study.

### Measurements

Before the examination, the subjects were asked to complete a questionnaire regarding their ordinary sleep using the PSQI Scale and lifestyle factors, including smoking habits, drinking habits and physical activity. The PSQI Scale included 19 self-administered questions that generated seven elements, including subjective sleep quality, sleep latency, sleep duration, habitual sleep efficiency, sleep disturbance, sleep medication use and daytime dysfunction. The score for each element ranged from 0 to 3, as follows: (1) subjective sleep quality (0=very good, 1=fairly good, 2=fairly bad and 3=very bad; (2) sleep latency, which was defined as the sum of the scores of (a) the required time in minutes to fall asleep each night (0 for ≤15 min, 1 for 16–30 min, 2 for 31–60 min, 3 for >60 min) and (b) the frequency of falling asleep within 30 min (0 for not during the past month, 1 for less than once a week, 2 for once or twice a week, 3 for three or more times a week), assigning a score of 0 for a sum of 0, 1 for a sum of 1–2, 2 for a sum of 3–4 and 3 for a sum of 5–6; (3) sleep duration (0 for >7 hours, 1 for 6–≤7 hours, 2 for 5–<6 hours and 3 for <5 hours); (4) habitual sleep efficiency, which was defined as the proportion of hours slept to hours spent in bed (0 for ≥85%, 1 for 75%–84%, 2 for 65%–74% and 3 for <65%); (5) frequency of sleep disturbance, which was defined as having trouble sleeping for some reason (0 for not during the past month, 1 for less than once a week, 2 for once or twice a week, 3 for three or more times a week), assigning 0 for a total score of 0, 1 for a total score of 1–9, 2 for a total score of 10–18 and 3 for a total score of 19–27; (6) sleep medication use (0 for no use during the past month, 1 for less than once a week, 2 for once or twice a week, and 3 for three or more times a week; and (7) frequency of daytime dysfunction, which was defined as the sum of the scores of (a) having trouble staying awake while driving, eating meals or engaging in social activity (0 for never, 1 for once or twice, 2 for once or twice each week, 3 for three or more times each week) and (b) having trouble in sustaining enough enthusiasm to get things done (0 for no problem at all, 1 for only a

very slight problem, 2 for somewhat of a problem, 3 for a serious problem), assigning 0 for a total score of 0, 1 for a total score of 1–2, 2 for a total score of 3–4 and 3 for a total score of 5–6. The cumulative score of these elements was defined as the global score, which ranged from 0 to 21. Poor sleepers were identified based on a global score of >5.

Physical activity was defined as walking or an equivalent amount of physical activity for more than 1 hour a day. The subjects were examined after an overnight fast. Ultrasound was used to evaluate hepatic steatosis. An ultrasound diagnosis of fatty liver was defined as a bright liver, with increased liver echogenicity compared with that in the kidneys, vascular blurring and deep attenuation of the liver.[16]

### Statistical analysis

All statistical analyses were conducted separately by sex, based on the fact that both sleep status and NAFLD have sex differences. The two groups (men and women) were compared using the Mann-Whitney U-test and the $\chi^2$ test for categorical variables. For continuous variables, analysis of covariance was performed to determine any significant differences between the non-NAFLD and NAFLD groups, after adjustments for age, smoking habits and physical exercise. After adjusting for the possible confounding factors, such as age, body mass index (BMI), smoking habits and physical activity, logistic regression analysis was conducted to assess the relationships of sleep disorder, based on the global PSQI score and the other seven elements, with NAFLD. The values for ORs with 95% CIs were calculated from the logistic regression analysis. Statistical analyses were performed using SPSS V.25.0 for Windows. A p value of <0.05 was considered significant.

## PATIENT AND PUBLIC INVOLVEMENT

Patients and public were not included in the present study. Patient consent is not required.

## RESULTS

### Characteristics of the participants

The participants' characteristics were compared by sex to clarify any sex differences (table 1). The mean age was significantly higher in women than in men (55.5 years vs 56.6 years; p=0.002). Compared with men, women had significantly lower BMI and sleep duration; lower proportions of subjects with a smoking history and NAFLD; higher level of physical activity and significantly higher global PSQI scores (p<0.001 for all).

### Comparison between non-NAFLD and NAFLD

The mean score for sleep medication use was significantly higher in those without NAFLD than in those with NAFLD in both men (0.21±0.74 vs 0.13±0.59; p=0.044) and women (0.28±0.82 vs 0.20±0.70; p=0.030) (table 2). In men, the mean scores for sleep latency (0.58±0.74 vs

| Table 1 | Characteristics of participants | | |
|---|---|---|---|
| | Men (n=1864) | Women (n=2964) | P value |
| Age (years) | 55.5±12.7 | 56.6±12.4 | 0.002 |
| Body mass index (kg/m$^2$) | 23.4±3.3 | 22.9±3.5 | <0.001 |
| Sleep duration (min/day) | 388.0±63.0 | 380.0±59.0 | <0.001 |
| Physical activity | 41.5% (774) | 47.1% (1395) | <0.001 |
| Current smoker | 25.1% (467) | 5.6% (165) | <0.001 |
| NAFLD | 47.0% (876) | 37.1% (1099) | <0.001 |
| ALT (U/L) | 24.5±15.5 | 20.7±15.2 | <0.001 |
| Global PSQI score | 4.9±2.6 | 5.2±2.6 | <0.001 |

ALT, alanine aminotransferase; NAFLD, non-alcoholic fatty liver disease; PSQI, Pittsburgh Sleep Quality Index.

0.63±0.78; p=0.066) and daytime dysfunction (0.62±0.72 vs 0.71±0.80; p=0.074) tended to be lower in those without NAFLD than in those with NAFLD.

### Association between sleep disorder and NAFLD

In men, the OR for NAFLD was lower with a score of 3 for sleep medication use (OR 0.60, 95% CI 0.38–0.95, p=0.030) than with a score of 0 (no sleep medication use) (table 3); in women, the tendency was similar, but the OR for NAFLD was not significant (OR 0.75, 95% CI 0.54–1.04, p=0.084) (table 4). After further adjustment for BMI, sleep medication use was not associated with NAFLD in both men and women.

The OR for NAFLD was higher with a daytime dysfunction score of 3 than a score of 0 in both men (OR 2.82, 95% CI 1.39–5.75, p=0.004) and women (OR 2.08, 95% CI 1.10–3.92, p=0.024). After further adjustments for BMI, daytime dysfunction was not associated with NAFLD in men, but it was in women with a daytime dysfunction score of 3 compared with those with a score of 0 (OR 2.62, 95% CI 1.20–5.72, p=0.015).

In men, the OR for NAFLD was higher with a score of 2 for sleep latency (OR 1.66, 95% CI 1.14–2.02, p=0.008) than with a score of 0 after adjustment for BMI. In women, the OR for NAFLD was lower with a score of 1 for sleep latency (OR 0.82, 95% CI 0.69–0.97, p=0.022) than with a score of 0. Moreover, the OR for NAFLD was higher with a score of 1 for habitual sleep efficiency (OR 1.29, 95% CI 1.09–1.53, p=0.004) than with a score of 0 in women. After further adjustments for BMI, sleep latency and habitual sleep efficiency were not associated with NAFLD in women.

The present results showed no associations of NAFLD with the global PSQI score, subjective sleep quality score, sleep duration score and sleep disturbance score.

### DISCUSSION

In the present study, the associations of NAFLD with all the elements of the PSQI were first evaluated, and it was found that the score for sleep medication use was

| Table 2 | Comparison of each PSQI score between the groups with and without NAFLD | | |
|---|---|---|---|
| | With NAFLD | Without NAFLD | P value |
| Men | | | |
| Global PSQI score | 5.02±2.65 | 4.85±2.47 | 0.216 |
| Subjective sleep quality | 1.14±0.61 | 1.07±0.59 | 0.077 |
| Sleep latency | 0.63±0.78 | 0.58±0.74 | 0.066 |
| Sleep duration | 1.31±0.81 | 1.27±0.82 | 0.863 |
| Habitual sleep efficiency | 0.40±0.66 | 0.40±0.65 | 0.808 |
| Sleep disturbance | 0.70±0.51 | 0.69±0.50 | 0.414 |
| Use of sleep medication | 0.13±0.59 | 0.21±0.74 | 0.044 |
| Daytime dysfunction | 0.71±0.80 | 0.62±0.72 | 0.074 |
| Women | | | |
| Global PSQI score | 5.19±2.59 | 5.29±2.64 | 0.409 |
| Subjective sleep quality | 1.11±0.60 | 1.11±0.59 | 0.935 |
| Sleep latency | 0.72±0.85 | 0.76±0.84 | 0.443 |
| Sleep duration | 1.40±0.77 | 1.41±0.78 | 0.314 |
| Habitual sleep efficiency | 0.40±0.63 | 0.40±0.70 | 0.672 |
| Sleep disturbance | 0.69±0.51 | 0.70±0.50 | 0.947 |
| Use of sleep medication | 0.20±0.70 | 0.28±0.82 | 0.030 |
| Daytime dysfunction | 0.66±0.73 | 0.63±0.69 | 0.390 |

NAFLD, non-alcoholic fatty liver disease; PSQI, Pittsburgh Sleep Quality Index.

**Table 3** OR of NAFLD by the PSQI and its components in men (n=1864)

| | Score | N | Model 1 | | Model 2 | |
|---|---|---|---|---|---|---|
| | | | OR (95% CI) | P value | OR (95% CI) | P value |
| Global PSQI score | ≤5 | 1202 | 1.00 (reference) | | 1.00 (reference) | |
| | ≥6 | 662 | 1.08 (0.89–1.30) | 0.458 | 1.17 (0.94–1.47) | 0.161 |
| Subjective sleep quality | 0 | 223 | 1.00 (reference) | | 1.00 (reference) | |
| | 1 | 1252 | 1.10 (0.83–1.47) | 0.506 | 1.17 (0.83–1.64) | 0.377 |
| | 2 | 363 | 1.24 (0.89–1.75) | 0.207 | 1.28 (0.86–1.90) | 0.224 |
| | 3 | 26 | 2.02 (0.87–4.67) | 0.101 | 1.68 (0.64–4.45) | 0.295 |
| Sleep latency | 0 | 1011 | 1.00 (reference) | | 1.00 (reference) | |
| | 1 | 624 | 1.05 (0.86–1.29) | 0.622 | 1.03 (0.82–1.31) | 0.781 |
| | 2 | 186 | 1.37 (1.00–1.88) | 0.050 | 1.66 (1.14–2.02) | 0.008 |
| | 3 | 43 | 1.31 (0.70–2.43) | 0.398 | 1.31 (0.63–2.74) | 0.467 |
| Sleep duration | 0 | 368 | 1.00 (reference) | | 1.00 (reference) | |
| | 1 | 656 | 0.99 (0.76–1.28) | 0.923 | 0.91 (0.67–1.23) | 0.537 |
| | 2 | 778 | 1.01 (0.78–1.31) | 0.937 | 0.89 (0.66–1.21) | 0.458 |
| | 3 | 62 | 1.04 (0.60–1.80) | 0.894 | 1.00 (0.52–1.93) | 0.999 |
| Habitual sleep efficiency | 0 | 1262 | 1.00 (reference) | | 1.00 (reference) | |
| | 1 | 500 | 1.00 (0.81–1.23) | 0.991 | 1.01 (0.79–1.29) | 0.931 |
| | 2 | 64 | 0.94 (0.57–1.57) | 0.820 | 0.79 (0.42–1.47) | 0.456 |
| | 3 | 38 | 1.23 (0.64–2.37) | 0.534 | 1.71 (0.82–3.55) | 0.152 |
| Sleep disturbance | 0 | 607 | 1.00 (reference) | | 1.00 (reference) | |
| | 1 | 1218 | 1.07 (0.88–1.30) | 0.499 | 1.08 (0.86–1.37) | 0.493 |
| | 2 | 38 | 1.13 (0.58–2.19) | 0.726 | 0.93 (0.41–2.10) | 0.852 |
| | 3 | 1 | – | – | – | – |
| Use of sleep medication | 0 | 1730 | 1.00 (reference) | | 1.00 (reference) | |
| | 1 | 26 | 0.97 (0.44–2.12) | 0.939 | 1.09 (0.46–2.62) | 0.842 |
| | 2 | 19 | 1.06 (0.43–2.65) | 0.894 | 1.40 (0.53–3.90) | 0.522 |
| | 3 | 89 | 0.60 (0.38–0.95) | 0.030 | 0.61 (0.36–1.04) | 0.068 |
| Daytime dysfunction | 0 | 912 | 1.00 (reference) | | 1.00 (reference) | |
| | 1 | 702 | 1.00 (0.82–1.22) | 0.992 | 1.16 (0.92–1.47) | 0.203 |
| | 2 | 210 | 1.11 (0.82–1.50) | 0.520 | 1.16 (0.81–1.66) | 0.414 |
| | 3 | 40 | 2.82 (1.39–5.75) | 0.004 | 2.04 (0.92–4.54) | 0.079 |

Model 1: adjustment for age, smoking habits and physical activity; model 2: model 1 plus adjustment for body mass index.
NAFLD, non-alcoholic fatty liver disease; PSQI, Pittsburgh Sleep Quality Index.

significantly higher in patients without NAFLD than in those with NAFLD. In addition, the score for sleep medication use in men was associated with a relatively low OR for NAFLD, whereas the score for daytime dysfunction was associated with a relatively high OR for NAFLD in both sexes. The scores for sleep latency in both sexes and habitual sleep efficiency in women were also associated with NAFLD.

Previous studies have suggested the underlying mechanisms by which sleep quality contributes to NAFLD. Poor sleep quality activates the hypothalamic–pituitary–adrenal (HPA) axis, thereby enhancing the secretion of stress hormones, such as cortisol and catecholamines, which can increase the risk for metabolic syndrome.[17–19]

Other studies have shown the association between poor sleep quality and hypothalamic orexin,[20 21] which controls the secretion of the appetite-regulating hormones leptin and ghrelin.[22] These hormones play roles in the regulation of glucose metabolism.[23] In fact, poor sleep quality was reported to increase insulin resistance in patients with type 2 diabetes mellitus.[24] Oestrogen activates the HPA axis response to stress,[25] and insulin resistance due to poor sleep quality develops only in women,[26] explaining possible sex differences in the relationship between sleep quality and NAFLD.

Although both short sleep duration and sleep quality have been associated with disease severity and insulin resistance in NAFLD,[12] the use of sleeping pills, regardless

**Table 4** OR of NAFLD by the PSQI and its components in women (n=2964)

| | Score | N | Model 1 | | Model 2 | |
|---|---|---|---|---|---|---|
| | | | OR (95% CI) | P value | OR (95% CI) | P value |
| Global PSQI score | ≤5 | 1783 | 1.00 (reference) | | 1.00 (reference) | |
| | ≥6 | 1181 | 0.99 (0.85–1.15) | 0.847 | 1.17 (0.94–1.47) | 0.161 |
| Subjective sleep quality | 0 | 332 | 1.00 (reference) | | 1.00 (reference) | |
| | 1 | 2029 | 0.86 (0.68–1.10) | 0.233 | 0.93 (0.70–1.23) | 0.610 |
| | 2 | 554 | 1.02 (0.77–1.35) | 0.902 | 1.11 (0.80–1.55) | 0.538 |
| | 3 | 49 | 0.67 (0.35–1.30) | 0.233 | 0.81 (0.39–1.68) | 0.565 |
| Sleep latency | 0 | 1399 | 1.00 (reference) | | 1.00 (reference) | |
| | 1 | 1038 | 0.82 (0.69–0.97) | 0.022 | 0.83 (0.68–1.01) | 0.068 |
| | 2 | 405 | 0.95 (0.75–1.20) | 0.659 | 0.98 (0.74–1.29) | 0.865 |
| | 3 | 122 | 1.02 (0.69–1.50) | 0.920 | 0.82 (0.52–1.29) | 0.392 |
| Sleep duration | 0 | 430 | 1.00 (reference) | | 1.00 (reference) | |
| | 1 | 994 | 1.15 (0.90–1.45) | 0.262 | 1.22 (0.92–1.60) | 0.169 |
| | 2 | 1444 | 0.94 (0.75–1.18) | 0.579 | 0.92 (0.70–1.20) | 0.520 |
| | 3 | 96 | 1.17 (0.74–1.84) | 0.510 | 1.07 (0.63–1.82) | 0.810 |
| Habitual sleep efficiency | 0 | 2030 | 1.00 (reference) | | 1.00 (reference) | |
| | 1 | 749 | 1.29 (1.09–1.53) | 0.004 | 1.21 (0.99–1.48) | 0.069 |
| | 2 | 118 | 0.80 (0.53–1.21) | 0.296 | 0.96 (0.60–1.54) | 0.859 |
| | 3 | 67 | 0.73 (0.42–1.26) | 0.255 | 0.66 (0.37–1.25) | 0.199 |
| Sleep disturbance | 0 | 956 | 1.00 (reference) | | 1.00 (reference) | |
| | 1 | 1946 | 0.97 (0.83–1.14) | 0.725 | 0.97 (0.80–1.17) | 0.763 |
| | 2 | 62 | 1.17 (0.69–1.99) | 0.551 | 1.30 (0.70–2.44) | 0.410 |
| | 3 | 0 | – | | | – |
| Use of sleep medication | 0 | 2644 | 1.00 (reference) | | 1.00 (reference) | |
| | 1 | 79 | 0.69 (0.42–1.13) | 0.142 | 0.97 (0.55–1.72) | 0.919 |
| | 2 | 54 | 0.68 (0.37–1.24) | 0.202 | 0.62 (0.31–1.22) | 0.166 |
| | 3 | 187 | 0.75 (0.54–1.04) | 0.084 | 0.69 (0.46–1.02) | 0.061 |
| Daytime dysfunction | 0 | 1417 | 1.00 (reference) | | 1.00 (reference) | |
| | 1 | 1234 | 0.96 (0.82–1.13) | 0.626 | 0.98 (0.81–1.19) | 0.848 |
| | 2 | 273 | 1.04 (0.79–1.36) | 0.805 | 0.94 (0.68–1.30) | 0.718 |
| | 3 | 40 | 2.08 (1.10–3.92) | 0.024 | 2.62 (1.20–5.72) | 0.015 |

Model 1: adjustment for age, smoking habits and physical activity; model 2: model 1 plus adjustment for body mass index.
NAFLD, non-alcoholic fatty liver disease; PSQI, Pittsburgh Sleep Quality Index.

of the drug type, has been shown to improve sleep quality and extend sleeping time.[27–31] On other hand, ramelteon, which is a melatonin receptor agonist, may be beneficial for NAFLD by repairing the circadian rhythm, which is associated with glucose metabolism and insulin sensitivity.[32] The effect of sleeping pills for preventing NAFLD may be worth examining.

Daytime dysfunction is a possible consequence of sleep disruption and may be directly reflected by daytime sleepiness. The present study supported the previous studies that reported the associations of daytime sleepiness with elevations of liver enzymes, insulin resistance and the histological features of NAFLD.[12] On the other hand, in the present study, there was no difference in the mean daytime dysfunction score between the non-NAFLD and NAFLD groups. In agreement with the present results, the results of a previous study showed that the median Epworth Sleepiness Scale did not differ significantly between the non-NAFLD and NAFLD groups.[12]

Obesity is a major risk factor for NAFLD. Therefore, scores of the PSQI except daytime dysfunction in women were not associated with NAFLD after adjustment for BMI. This suggests that the effects of sleep quality on NAFLD are mediated by obesity because of the strong association between obesity and sleep quality.[33] On the other hand, the prevalence of NAFLD is much higher than would be expected from a population with similar BMI in the present study. This may be explained by the fact

that subjects in the present study often had NAFLD with normal or low body weight, and ultrasound has a high sensitivity for detection of fatty liver in such persons. The high proportion of lean NAFLD in Japan compared with western countries may support the above interpretation.[34]

Although this study had the strength of a large sample size from multiple centres, it had some limitations. First, being a self-reported questionnaire, the PSQI assessment was subjective. Therefore, the associations between sleep quality and NAFLD need to be confirmed by objective methods, such as polysomnography and actigraphy. Second, this study was cross-sectional and evaluated sleep characteristics at the same time as NAFLD. Therefore, further longitudinal evaluation would be essential to elucidate the causal relationship between sleep quality and NAFLD. Third, the specific names of the sleep medications being taken by the patients were not identified. In the future, it would be interesting to know which sleep medication has the most favourable effect on preventing NAFLD. Fourth, some of the results of this study on the association between sleep quality and NAFLD were contrary to those in previous reports.[9 12 13] This may be explained by the fact that sleep quality could be affected by many factors, such as cultural, physical, psychological and ethnic differences.[35] Fifth, the degree of fatty liver was not evaluated by semiquantitative ultrasonographic indices in the present study. Future studies should explore the association between sleep quality and the degree of fat or fibrosis in NAFLD.

## CONCLUSIONS

This study demonstrated that sleep medication use and daytime dysfunction were associated with NAFLD. These findings suggest that additional studies are needed to assess the importance, if any, of sleeping pills for the treatment of NAFLD.

**Contributors** AT, NT, and HO contributed to the design of the present study. AT, MK, YA, MK, YK and TO were responsible for data collection and overseeing the study procedures. The analysis was conducted by AT. The manuscript was written by AT and HO, and YA, MK, MK, YK, TO, MF, MH, HI, KA and NT made contributions to the interpretation of the results in present study. All authors read and approved the final version of the manuscript.

**Funding** The authors have not declared a specific grant for this research from any funding agency in the public, commercial or not-for-profit sectors.

**Competing interests** None declared.

**Patient and public involvement** Patients and/or the public were not involved in the design, or conduct, or reporting, or dissemination plans of this research.

**Patient consent for publication** Obtained.

**Ethics approval** The study protocol was approved by the ethics committee of Fukushima Medical University School of Medicine (#1636). All patients provided written, informed consent prior to study participation, and this study was conducted in conformity with the ethical guidelines of the Declaration of Helsinki.

**Provenance and peer review** Not commissioned; externally peer reviewed.

**Data availability statement** No data are available. We do not have permission from the participants to share this data in whole or in part.

**ORCID iD**
Atsushi Takahashi http://orcid.org/0000-0003-0568-8361

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
