## [Reviewer comments · BMJ Open]

ARTICLE DETAILS

TITLE (PROVISIONAL)	Effects of sleep quality on non-alcoholic fatty liver disease: A cross-sectional survey
AUTHORS	Takahashi, Atsushi; Anzai, Yukio; Kuroda, Masahito; Kokubun, Masae; Kondo, Yuichiro; Ogata, Takashi; Fujita, Masashi; Hayashi, Manabu; Imaizumi, Hiromichi; Abe, Kazumichi; Tanji, Nobuo; Ohira, Hiromasa

VERSION 1 – REVIEW

REVIEWER	Amedeo Lonardo Azienda Ospedaliero Universitaria, Modena Italy
REVIEW RETURNED	14-May-2020

GENERAL COMMENTS	GENERAL COMMENT One point of strength of the present study is the evaluation of sex differences in regard to disrupted sleep physiology in NAFLD. Authors may be willing to further develop a) the rationale for exploring sex differences [Background] in NAFLD and b) to fully discuss the potential physiopathological grounds underlying such differences [Discussion]. In addition, diagnostic criteria are poor and must be improved. SPECIFIC COMMENT MAJOR “Ultrasound was used to evaluate hepatic steatosis. An ultrasound diagnosis of fatty liver was defined as a bright liver; increased liver echogenicity, compared with that in the kidneys; vascular blurring; and deep attenuation of the liver” □ Please use a supporting reference. Moreover, these authors should also define which type of fatty liver was this ? Alcoholic ? Nonalcoholic ? Viral ? Among the limitations of the study, please discuss failure to use semi-quantitative ultrasonographic indices which predict liver histology and metabolic derangements. The discussion must be reworked. Delete the following statement “Lifestyle is an important cause of metabolic diseases, such as NAFLD. However, compared with nutrition or exercise, sleep disorder has gained less attention. Among the several sleep disorders, OSA and short sleep duration have been established as risk factors for NAFLD [6-13]. On the other hand, the detailed effects of sleep quality on the risk for NAFLD remain unclear.” These comments may be utilized in the Introduction. Given the multiple limitations in the design of this study the following sentence must be reworked and downplayed “the use of
--

	sleeping pills could be the new goals and cutting edge treatment for NAFLD.” For example I would suggest “additional studies are awaited to assess the importance, if any, of sleeping pills.....” MINOR The section on Strengths and limitations of this study is written in poor English and must carefully be edited.
--	---

REVIEWER	Jieli Lu Shanghai Institute of Endocrine and Metabolic Diseases, Ruijin Hospital, Shanghai Jiao Tong University School of Medicine, China.
REVIEW RETURNED	22-Jun-2020

GENERAL COMMENTS	Based on a large sample size from multiple centers, the authors examined the association between sleep quality, based on the Pittsburgh Sleep Quality Index, and non-alcoholic fatty liver disease. The topic is important and interesting. I have some concerns regarding the description.  1. The major limitation of the present study is that both sleep characters are measured at the same moment in time as NAFLD. In the present case, sleep medication use and sleep quality may be affected by the diagnosis of NAFLD. 2. I confused why the authors compared the characteristics of men and women in table 1. The comparison of NAFLD and non-NAFLD at first could be more important. 3. In this study, the authors did a sex-specific analysis. They concluded that “Sleep medication use and daytime dysfunction were associated with NAFLD”. However, sleep medication use was not associated with NAFLD in women in the table 4. 4. Please clarify the sex heterogeneity more clearly. How do you think about the difference in men and women? Please discuss it in the manuscript. 5. We are told that the score for each element ranged from 0 to 3. But it was not clear about the accurate division of the score of sleep latency, frequency of sleep disturbance, and the frequency of daytime dysfunction. 6. Please define the diagnostic method of NAFLD more clearly. 7. The authors did not clearly describe the variables adjusted in the logistic regression in statistical analysis, the results section, and tables. Did the current results adjust BMI and ALT? 8. Metabolic indicators such as lipids concentration and socioeconomic factors are also confounders. 9. The introduction of the previous literature was not clear. Lack of updated references. Besides, it is unclear how much the current research contributes to this topic. Please don't talk about obstructive sleep apnea syndrome frequently. The discussion was really confused and disordered.
--

REVIEWER	Ilaria Umbro Sapienza University of Rome, Italy
REVIEW RETURNED	27-Jul-2020

GENERAL COMMENTS	Dear Authors, I reviewed the manuscript number ID bmjopen-2020-039947 entitled "Influence of sleep quality on non-alcoholic fatty liver
---

	disease : a cross-sectional survey" with Dr. Atsushi Takahashi as correspondence author. Recently, the association between sleep medicine and gastroenterology attracts great attention in literature. In this study Authors "aimed to clarify the detailed association between sleep quality and NAFLD by analyzing all elements of the PSQI separately by sex and by adjusting for confounding factors." I think Authors should remove "detailed" from the aim because it could be confusing. I appreciate that Authors reported as a limitation of the study that, since it is based on a self-reported questionnaire (Pittsburgh Sleep Quality Index), needs to be confirmed by objective methods. On the contrary, the "detailed association" may seem that the study is based on objective methods. ABSTRACT Authors should better describe their results because seem to contradict the conclusion. INTRODUCTION Pag 5 of 24, lines 15-25: "Recently, the association of sleep disorder with NAFLD has attracted attention. Obstructive sleep apnea (OSA) and short sleep duration have been known to be associated with NAFLD [6-8], but only few have reported on the association between sleep quality and NAFLD [8-10]." Since Authors stated "recently" I suggest to update the references. Please see "Umbro I, Fabiani V, Fabiani M, Angelico F, Del Ben M. Association between non-alcoholic fatty liver disease and obstructive sleep apnea. World J Gastroenterol. 2020 May 28;26(20):2669-2681. doi: 10.3748/wjg.v26.i20.2669. PMID: 32523319; PMCID: PMC7265151." RESULTS Authors should remove the brackets and their contents from the subtitles. DISCUSSION Pag 15 of 24, lines 36-43: "The results of this study suggested that sleeping pills may be a new treatment or preventive strategy for NAFLD in patients with sleep disorder." From this study, I think it is not quite right to state this. Authors should restate their results and Conclusion section as a consequence. I think the manuscript should be revised before publication in BMJ Open.
--	---

VERSION 1 – AUTHOR RESPONSE

Reviewer: 1
Reviewer Name
Amedeo Lonardo
Institution and Country
Azienda Ospedaliero Universitaria, Modena Italy

GENERAL COMMENT

One point of strength of the present study is the evaluation of sex differences in regard to disrupted sleep physiology in NAFLD. Authors may be willing to further develop a) the rationale for exploring sex differences [Background] in NAFLD and b) to fully discuss the potential physiopathological grounds underlying such differences [Discussion]. In addition, diagnostic criteria are poor and must be improved.

→There are sex differences in sleeping time and distribution of sleeping time in patients with NAFLD, as previously reported. We referred to this in the introduction (page 5, lines 2-3) and discussion (page 15, lines 10-12), as suggested. Moreover, the diagnostic criteria were improved (page 5, lines 14-16).

SPECIFIC COMMENT

MAJOR

“Ultrasound was used to evaluate hepatic steatosis. An ultrasound diagnosis of fatty liver was defined as a bright liver; increased liver echogenicity, compared with that in the kidneys; vascular blurring; and deep attenuation of the liver” □ Please use a supporting reference. Moreover, these authors should also define which type of fatty liver was this ? Alcoholic ? Nonalcoholic ? Viral ?

→We added an appropriate reference, as suggested (reference 16). We ruled out fatty liver due to alcohol or viral causes. (page 5, lines 14-16)

Among the limitations of the study, please discuss failure to use semi-quantitative ultrasonographic indices which predict liver histology and metabolic derangements.

→We added this as a limitation, as suggested. (page 17, lines 14-17)

The discussion must be reworked. Delete the following statement “Lifestyle is an important cause of metabolic diseases, such as NAFLD. However, compared with nutrition or exercise, sleep disorder has gained less attention. Among the several sleep disorders, OSA and short sleep duration have been established as risk factors for NAFLD [6-13]. On the other hand, the detailed effects of sleep quality on the risk for NAFLD remain unclear.” These comments may be utilized in the Introduction.

→We made the suggested changes. (page 4, lines 9-11)

Given the multiple limitations in the design of this study the following sentence must be reworked and downplayed “the use of sleeping pills could be the new goals and cutting edge treatment for NAFLD.” For example I would suggest “additional studies are awaited to assess the importance, if any, of sleeping pills.....”

→We revised the sentence appropriately. (page 18, lines 3-4)

MINOR

The section on Strengths and limitations of this study is written in poor English and must carefully be edited.

→A native English-speaking medical editor edited the paper for us.

Reviewer: 2

Reviewer Name

Jieli Lu

Institution and Country

Shanghai Institute of Endocrine and Metabolic Diseases, Ruijin Hospital, Shanghai Jiao Tong University School of Medicine, China.

Please leave your comments for the authors below

Based on a large sample size from multiple centers, the authors examined the association between sleep quality, based on the Pittsburgh Sleep Quality Index, and non-alcoholic fatty liver disease. The topic is important and interesting. I have some concerns regarding the description.

1. The major limitation of the present study is that both sleep characters are measured at the same moment in time as NAFLD. In the present case, sleep medication use and sleep quality may be affected by the diagnosis of NAFLD.

→We agree with the possibility of interaction between sleep quality and the diagnosis of NAFLD and refer to it in the limitations. (page 17, lines 5-8)

2. I confused why the authors compared the characteristics of men and women in table 1. The comparison of NAFLD and non-NAFLD at first could be more important.

→As mentioned by reviewer 1, one strength of the present study was the evaluation of sex differences in disrupted sleep physiology in NAFLD. Therefore, we compared the differences in characteristics by sex and mentioned it in the results.

3. In this study, the authors did a sex-specific analysis. They concluded that “Sleep medication use and daytime dysfunction were associated with NAFLD”. However, sleep medication use was not associated with NAFLD in women in the table 4.

→Based on the comments, we changed it as follows: Sleep quality was associated with NAFLD, and there were sex differences. (page 3, line 2)

4. Please clarify the sex heterogeneity more clearly. How do you think about the difference in men and women? Please discuss it in the manuscript.

→We refer to the sex differences in the introduction (page 5, lines 2-3) and discussion sections (page 15, lines 10-12).

5. We are told that the score for each element ranged from 0 to 3. But it was not clear about the accurate division of the score of sleep latency, frequency of sleep disturbance, and the frequency of daytime dysfunction.

→We noted that the above score ranged from 0 to 3 in the measurements (page 6, line 13- page 7, line 17).

6. Please define the diagnostic method of NAFLD more clearly.

→We clarified the definition of NAFLD. (page 5, lines 14-16)

7. The authors did not clearly describe the variables adjusted in the logistic regression in statistical analysis, the results section, and tables. Did the current results adjust BMI and ALT?

→We adjusted for BMI and added it to the table as Model 2.

8. Metabolic indicators such as lipids concentration and socioeconomic factors are also confounders.

→We agree with the comment. In this study, we adjusted as much as possible for confounders.

9. The introduction of the previous literature was not clear. Lack of updated references. Besides, it is unclear how much the current research contributes to this topic. Please don't talk about obstructive sleep apnea syndrome frequently. The discussion was really confused and disordered.

→We referred to obstructive sleep apnea only in the introduction.

Reviewer: 3

Reviewer Name

Omar Mesarwi

Institution and Country

University of California, San Diego, U.S.A.

In this study by Takahashi et al., the authors examine the link between sleep quality indices as gauged by the PSQI, and NAFLD presence. This is an interesting line of research as the associations between sleep and metabolism deserve further exploration. However, there are some issues with the data as reported:

1. The main issue is that the authors have not identified cause and effect, but appear to make claims suggesting this. The significant finding in this paper is that sleep medication use was higher in NAFLD, and the link is associative. The authors write that "the higher score for sleep medication use in non-NAFLD than in NAFLD implied the favorable effects of sleeping pills on metabolic factors. The results of this study suggested that sleeping pills may be a new treatment or preventive strategy for NAFLD in patients with sleep disorder." This is speculative and is not implied by the current study. The authors also state: "Our data revealed both the problems and the solutions for sleep disorder in NAFLD." This is again entirely too generous an interpretation of the available data. Same with: "These findings suggested that improvement of daytime dysfunction and the use of sleeping pills could be the new goals and cutting edge treatment for NAFLD."

→We deleted the above descriptions, as suggested.

2. Sleep medication use is known to correlate with age and BMI but the effects of these as confounding variables is not considered.

→We adjusted for BMI and added it to the table as Model 2.

3. The prevalence of NAFLD is much higher than would be expected from a population with similar BMI and this should be discussed. Was there selection bias in the study?

→We added the description below based on the comments.

Obesity is a major risk factor for NAFLD. Therefore, scores of the PSQI except daytime dysfunction in women were not associated with NAFLD after adjustment for BMI. This suggests that the effects of sleep quality on NAFLD are mediated by obesity because of the strong association between obesity and sleep quality [33]. On the other hand, the prevalence of NAFLD is much higher than would be expected from a population with similar BMI in the present study. This may be explained by the fact that subjects in the present study often had NAFLD with normal or low body weight, and ultrasound

has a high sensitivity for detection of fatty liver in such persons. The high proportion of lean NAFLD in Japan compared to western countries may support the above interpretation [34]. (page 16, line10- page17, line 1)

4. There is very possibly false discovery in table 4 as presented. For example, effect of sleep latency and sleep efficiency score of 1 is statistically significant, but not 2 or 3. With so many comparisons this is unlikely to be relevant. It's also not plausible that these should go in opposite directions with respect to the odds of having NAFLD.

→We adjusted for BMI as Model 2. We could not rule out an association between NAFLD and sleep latency or sleep efficiency, because similar opposite directions in the odds of metabolic syndromes were reported in a previous study (Okubo et al, BMC Public Health 2014, 14:562).

Reviewer: 4

Reviewer Name

Ilaria Umbro

Institution and Country

Sapienza University of Rome, Italy

Dear Authors,

I reviewed the manuscript number ID bmjopen-2020-039947 entitled "Influence of sleep quality on non-alcoholic fatty liver disease : a cross-sectional survey" with Dr. Atsushi Takahashi as correspondence author.

Recently, the association between sleep medicine and gastroenterology attracts great attention in literature. In this study Authors "aimed to clarify the detailed association between sleep quality and NAFLD by analyzing all elements of the PSQI separately by sex and by adjusting for confounding factors."

I think Authors should remove "detailed" from the aim because it could be confusing. I appreciate that Authors reported as a limitation of the study that, since it is based on a self-reported questionnaire (Pittsburgh Sleep Quality Index), needs to be confirmed by objective methods. On the contrary, the "detailed association" may seem that the study is based on objective methods.

→We removed "detailed" from the aim, as suggested.

ABSTRACT

Authors should better describe their results because seem to contradict the conclusion.

→We revised the conclusion according to the comments as follows: Sleep quality was associated with NAFLD, and there were sex differences. (page 3, line 2)

INTRODUCTION

Pag 5 of 24, lines 15-25: "Recently, the association of sleep disorder with NAFLD has attracted attention. Obstructive sleep apnea (OSA) and short sleep duration have been known to be associated with NAFLD [6-8], but only few have reported on the association between sleep quality and NAFLD [8-10]." Since Authors stated "recently" I suggest to update the references. Please see "Umbro I, Fabiani V, Fabiani M, Angelico F, Del Ben M. Association between non-alcoholic fatty liver disease and obstructive sleep apnea. World J Gastroenterol. 2020 May 28;26(20):2669-2681. doi: 10.3748/wjg.v26.i20.2669. PMID: 32523319; PMCID: PMC7265151."

→We added the above paper in the introduction. (reference 7)

RESULTS

Authors should remove the brackets and their contents from the subtitles.

→We made the suggested changes.

DISCUSSION

Pag 15 of 24, lines 36-43: "The results of this study suggested that sleeping pills may be a new treatment or preventive strategy for NAFLD in patients with sleep disorder."

From this study, I think it is not quite right to state this. Authors should restate their results and Conclusion section as a consequence.

→We revised our paper as follows: These findings suggest that additional studies are needed to assess the importance, if any, of sleeping pills for the treatment of NAFLD. (page 18, lines 3-4)

VERSION 2 – REVIEW

REVIEWER	Jieli Lu Shanghai Institute of Endocrine and Metabolic Diseases, Ruijin Hospital, Shanghai Jiao Tong University School of Medicine, China
REVIEW RETURNED	20-Aug-2020

GENERAL COMMENTS	Because of the cross-sectional nature of this study, the authors cannot conclude a causal association between sleep quality and NAFLD. However, the topic itself is important and interesting. Also, they have addressed the comments with point-by-point responses, and revised the manuscript accordingly. They found that sleep quality was associated with NAFLD, and there were sex differences. Thus, I recommended to publish this article and look forward to this team focusing on this topic, researching the longitudinal association between sleep quality and NAFLD in the further.
--

REVIEWER	Ilaria Umbro Sapienza University of Rome, Italy
REVIEW RETURNED	26-Aug-2020

GENERAL COMMENTS	Dear Authors, I reviewed the manuscript ID bmjopen-2020-039947.R1 entitled "Effects of sleep quality on non-alcoholic fatty liver disease: A cross-sectional survey" with Dr Atsushi Takahashi as correspondence author.
---

	Authors correctly modified the manuscript according to my revisions. Nevertheless, there are two further minor revisions: 1- Table 1 (page 11 of 52, line 57), Authors should replace “388 ± 63” with “388.0 ± 63.0” and “380 ± 59” with “380.0 ± 59.0”. 2- Table 4 (page 16 of 52, lines 12-13), Authors should replace “0.804” with “0.80” and “0.726” with “0.73”. On the basis of the ongoing effort of BMJ Open to increase its impact, this manuscript can be accepted after these revisions.
--	--